# An Italian Multicenter Perspective Harmonization Trial for the Assessment of MET Exon 14 Skipping Mutations in Standard Reference Samples

**DOI:** 10.3390/diagnostics13040629

**Published:** 2023-02-08

**Authors:** Paolo Bironzo, Francesco Pepe, Gianluca Russo, Pasquale Pisapia, Gianluca Gragnano, Gabriella Aquino, Silvia Bessi, Simonetta Buglioni, Federico Bartoccini, Giuseppina Ferrero, Michela Anna Bresciani, Paola Francia di Celle, Francesca Sibona, Andrea Giusti, Alessandra Movilia, Renata Mariella Farioli, Alessandra Santoro, Domenico Salemi, Stefania Scarpino, Dino Galafate, Stefania Tommasi, Rosanna Lacalamita, Davide Seminati, Elham Sajjadi, Silvia Novello, Fabio Pagni, Giancarlo Troncone, Umberto Malapelle

**Affiliations:** 1Department of Oncology, S. Luigi Gonzaga Hospital, University of Turin, 10043 Orbassano, Italy; 2Department of Public Health, University Federico II of Naples, 80131 Naples, Italy; 3Department of Pulmonary Oncology, AORN Dei Colli Monaldi, 80131 Naples, Italy; 4Departmental Structure of Oncological Molecular Pathology, Oncological Department Azienda USL Toscana Centro, S. Stefano Hospital, 59100 Prato, Italy; 5Pathology Unit, IRCCS Regina Elena National Cancer Institute, 00144 Rome, Italy; 6Department of Pathology, ASST Cremona, 26100 Cremona, Italy; 7Molecular Pathology, AOU Città della Salute e della Scienza di Torino-Presidio Ospedaliero Molinette, 10126 Turin, Italy; 8ASL Toscana Nord Ovest, Pathology Unit, Centro Polispecialistico “Achille Sicari”, 54033 Carrara, Italy; 9Department of Pathology, ASST Ovest Milanese, Ospedale di Legnano, 20025 Legnano, Italy; 10Division of Hematology and Bone Marrow Transplantation, Ospedali Riuniti Villa Sofia-Cervello, 90146 Palermo, Italy; 11Pathology Unit, Department of Clinical and Molecular Medicine, St. Andrea University Hospital, University of Rome La Sapienza, 00189 Rome, Italy; 12Molecular Genetics Laboratory, IRCCS Istituto Tumori Giovanni Paolo II, 70124 Bari, Italy; 13Department of Surgery and Translational Medicine, Section of Pathology, Università degli Studi di Mila-no-Bicocca, 20126 Milan, Italy; 14Department of Oncology and Hemato-Oncology, University of Milan, 20136 Milan, Italy

**Keywords:** MET, NSCLC, molecular test

## Abstract

Lung cancer remains the leading cause of cancer deaths worldwide. International societies have promoted the molecular analysis of MET proto-oncogene, receptor tyrosine kinase (*MET*) exon 14 skipping for the clinical stratification of non-small cell lung cancer (NSCLC) patients. Different technical approaches are available to detect *MET* exon 14 skipping in routine practice. Here, the technical performance and reproducibility of testing strategies for *MET* exon 14 skipping carried out in various centers were evaluated. In this retrospective study, each institution received a set (*n* = 10) of a customized artificial formalin-fixed paraffin-embedded (FFPE) cell line (Custom *MET*ex14 skipping FFPE block) that harbored the *MET* exon 14 skipping mutation (Seracare Life Sciences, Milford, MA, USA), which was previously validated by the Predictive Molecular Pathology Laboratory at the University of Naples Federico II. Each participating institution managed the reference slides according to their internal routine workflow. *MET* exon 14 skipping was successfully detected by all participating institutions. Molecular analysis highlighted a median Cq cut off of 29.3 (ranging from 27.1 to 30.7) and 2514 (ranging from 160 to 7526) read counts for real-time polymerase chain reaction (RT-PCR) and NGS-based analyses, respectively. Artificial reference slides were a valid tool to harmonize technical workflows in the evaluation of *MET* exon 14 skipping molecular alterations in routine practice.

## 1. Introduction

Lung cancer still represents the leading cause of cancer deaths worldwide [1]. Remarkably, giant strides have been made in terms of the clinical management of lung cancer patients in recent years. In particular, the identification of different biomarkers able to predict responsiveness to target treatments has significantly improved progression-free survival (PFS), overall survival (OS), and quality of life for advanced stage non-small cell lung cancer (NSCLC) patients [2,3,4,5,6].

Among the different genomic alterations, including point mutations, insertions/deletions (indels), and gene fusions, a novel class of molecular aberrations should be considered. In this setting, MET proto-oncogene, receptor tyrosine kinase (MET) exon 14 skipping mutations play a key role in the management of advanced stage NSCLC patients [7]. As a general rule, point mutations, indels, or large-scale whole-exon deletions can be associated with a splice-site alteration with the subsequent loss of transcription of exon 14. This phenomenon determines the loss of the MET binding site for Y1003 CBL (an E3 ubiquitin ligase) in the juxtamembrane domain, with a reduction in MET ubiquitination, degradation, and an increase in signal transduction [8,9].

Overall, MET exon 14 skipping molecular alterations occur in approximately 3–4% of advanced stage NSCLC patients [9,10,11,12]. The increasing attention on MET exon 14 skipping is derived from the approval of two novel tyrosine kinase inhibitors (TKIs), namely capmatinib and tepotinib, for advanced stage NSCLC patients harboring this type of genomic alteration [12,13].

Due to the heterogeneous nature of the alterations that can lead to the development of MET exon 14 skipping, the different approaches that can be adopted to identify these molecular events remain an open issue. Among these, RNA- or DNA-based next-generation sequencing (NGS), anchored multiplex polymerase chain reaction (PCR), or real-time reverse transcriptase-polymerase chain reaction (real-time RT-PCR) methodologies are routinely employed by different laboratories to detect MET exon 14 skipping mutations [14,15,16].

Overall, in order to increase the identification of MET exon 14 skipping molecular alterations, and to not leave any patient behind, it is crucial to evaluate inter-laboratory concordance even when different molecular testing platforms are employed. In this setting, the adoption of cell lines with known genomic backgrounds may be a useful tool, as previously demonstrated [17,18,19].

In this study, we evaluated the concordance among laboratories with high molecular expertise adopting different molecular testing platforms, using artificial reference standards for the detection of the MET exon 14 skipping molecular alteration.

## 2. Materials and Methods

### 2.1. Study Design

This study aimed to evaluate inter-laboratory reproducibility for the analysis of MET exon 14 skipping on a series of standard reference samples distributed to Italian referral institutions for biomarker testing. Each participating center received a customized, artificial, formalin-fixed paraffin-embedded (FFPE) cell line (Custom METex14 skipping FFPE block) that harbored MET exon 14 skipping developed by Seracare Life Sciences (Milford, MA, USA). The artificial control was previously analyzed by the Predictive Molecular Pathology Laboratory at the University of Naples Federico II in order to internally validate the samples before shipping them to other labs. A total of *n* = 10 slides were sent to *n* = 10 referral laboratories for predictive molecular pathology testing. At each institution, the slides were managed following the center’s internal workflow. After removing the paraffin, the RNA was purified and stored according to the internal testing strategy. The quantity of nucleic acids and, when technically available, the nucleic acids fragmentation index were evaluated. Nucleic acids were analyzed by adopting the routine diagnostic procedures for MET exon 14 skipping of each institution. Within *n* = 30 working days, the adopted technical assays and all detected molecular alterations (including/Cq/reads count) were listed in a dedicated database and shared with the coordinator center (Figure 1).

### 2.2. Validation of Customized Reference Standard Sample

The Predictive Molecular Pathology Unit (University of Naples Federico II) received the customized artificial cell block (Custom METex14 skipping FFPE block) to evaluate MET exon 14 skipping mutations with its internal workflow before shipment to other institutions. Briefly, *n* = 6 slides from the FFPE sample were de-paraffinized and incubated overnight with Proteinase K (Qiagen, Hilden, Germany). Then, the RNA was extracted and purified using the AllPrep DNA/RNA mini kit (Qiagen, Hilden, Germany), according to the manufacturer’s instructions. Moreover, an additional haematoxylin/eosin (H/E) stained slide was digitalized by NanoZoomer 2.0 RS (Hamamatsu Photonics, Sunayama-cho, Naka-ku, Hamamatsu City, Shizuoka, Japan) and archived. RNA quantification and qualification were carried out on the TapeStation 4200 system (Agilent Technologies, Santa Clara, CA, USA) using the High Sensitivity RNA Assay on RNA ScreenTape (Agilent Technologies), as previously described [19]. Data analysis was carried out on proprietary Tape Station 4200 analysis software (Agilent Technologies). RNA quantity and the corresponding integrity number (RIN) were used to evaluate the RNA fragmentation profile, which was annotated. Molecular analysis was performed using the Easy^®^PGX ready ALK/ROS/RET/MET assay (Diatech Pharmacogenetics S.R.L., Jesi, Italy) on the Easy^®^PGX system, following standardized procedures. 

### 2.3. Technical Approaches

Overall, the nucleic acids were purified from the reference slides using manual and automated procedures (*n* = 6/10 (60.0%) and *n* = 4/10 (40.0%), respectively). For the manual procedures, the RNeasy FFPE kit (Qiagen) assay was adopted by 4 out 6 (66.6%) institutions, while the High Pure miRNA Isolation kit (Roche Diagnostics, Basel, Switzerland) and the MagMAX™ FFPE DNA/RNA Ultra Kit (Thermo Fisher Scientific, Waltham, MA, USA) were each employed by a single institution, respectively (16.7%). For those utilizing automated procedures, the MagCore Total RNA FFPE One-step kit (RBC Biosciences; Taiwan, New Taipei City) with the MagCore Super instrument (RBC Biosciences; Taiwan, New Taipei City) and the Maxwell^®^ RSC RNA FFPE Kit (Promega Corporation, Madison, WI, USA) with the Maxwell^®^ RSC instrument (Promega Corporation; Madison, WI, USA) were equally used (*n* = 2/4, 50% and *n* = 2/4, 50%). Among those using manual procedures, 3 out of 6 (50.0%) centers utilized the Qubit RNA HS Assay Kit (Thermo Fisher Scientific) with the Qubit fluorometer (Thermo Fisher Scientific), 2 out of 6 (33.3%) centers used the NanoDrop™ One/OneC Microvolume UV-Vis Spectrophotometer (Thermo Fisher Scientific), and 1 out 6 (16.7%) centers applied the Ion Library Quantitation kit (Thermo Fisher Scientific) with the QuantStudio 5 Real-Time PCR System (Thermo Fisher Scientific). Moreover, the NanoPhotometer N60 system (Implen, Westlake Village, CA, USA), NanoDrop™ One/OneC Microvolume UV-Vis Spectrophotometer (Thermo Fisher Scientific), and Quantifluor RNA system (Promega Corporation, Madison, WI, USA) were each adopted by a single institution, respectively, while RNA was not evaluated before molecular analysis in a single case (Table 1).

For the MET exon 14 testing strategy, RNA-based approaches were used in all instances. The RT-PCR-based approach and NGS platforms were equally adopted by receiving institutions (5 out of 10, respectively). In detail, all RT-qPCR-based institutions employed the Easy^®^PGX ready ALK/ROS/RET/MET assay (Diatech Pharmacogenetics S.R.L.) with an Easy^®^PGX system; in a single case, MET exon 14 skipping was also confirmed using the Idylla ™ GeneFusion assay (Biocartis, Jersey City, NJ, USA) on the Idylla™ platform. Moreover, in 2 out of 5 cases (40.0%), the Oncomine Focus Assay (Thermo Fisher Scientific) with the Io S5™ (center X) and Ion S5™ XL (center X) platforms and the Myriapod^®^ NGS Cancer Panel RNA assay (Diatech Pharmacogenetics S.R.L.) with the iSeq™ and MiSeq™ platforms (Illumina, San Diego, CA, USA) were used, respectively. In addition, the Oncomine™ Precision Assay GX (Thermo Fisher Scientific, Waltham, MA, USA) with the fully automated Ion Torrent Genexus System (Thermo Fisher Scientific, Waltham, MA, USA) was used by a single institution. Ion Reporter 5.18.4.0 software was used for variant inspection on samples processed with the Oncomine Focus Assay. In the remaining cases, proprietary analysis software was adopted for data analysis. 

## 3. Results

### 3.1. Validation of Customized Reference Standard Sample

After the deparaffinization protocol, RNA was directly extracted from the selected slides. In addition, to check the quantity of RNA, two sets of slides representative of the first and last cell-block sections were analyzed in order to verify the neoplastic cell content during FFPE processing. Briefly, RNA was quantified and qualified on both slide sections. Quite similar RNA amounts of 44.6 and 31.0 ng/µL were obtained from the first and second slide set, respectively. In addition, matched RIN values of 2.8 and 2.6 were obtained (Figure 2). 

Molecular analysis was successfully performed according to the manual instructions of the Easy^®^PGX ready ALK/ROS/RET/MET assay (Diatech Pharmacogenetics S.R.L., Jesi, Italy). Data analysis was carried out on the proprietary software of the Easy^®^PGX platform. A positive result for MET exon 14 skipping in each slide set was reported, as shown in Figure 3.

### 3.2. Ring Trial Results

All participating institutions were able to submit their findings prior to the cutoff date. Overall, a median RNA concentration of 24.4 ng/µL (ranging from 4.3 to 51.3 ng/µL) was detected. Particularly, 32.8 ng/µL (ranging from 18.0 to 51.3 ng/µL) and 7.7 ng/µL (ranging from 4.3 to 10.0 ng/µL) RNA were identified in manually and automated nucleic acid extraction workflows, respectively.

In all instances, MET exon 14 skipping was successfully identified. Accordingly, molecular analysis showed a median Cq cut off of 30.2 (ranging from 27.1 to 31.4) and 2514 (ranging from 160 to 7526) read counts for RT-qPCR- and NGS-based procedures, respectively. Moreover, the Cq and read count were not reported in a single case, respectively (Table 1).

## 4. Discussion

This study aimed to evaluate the technical feasibility of analyzing MET exon 14 skipping on customized artificial samples using different genomic platforms. Particularly, this multicenter observational study tried to highlight the robustness of technical workflows adopted by each participating center in the detection of the aberrant MET transcript for targeted therapy of selected NSCLC patients [7,9]. Preanalytical steps of sample management play a pivotal role in the molecular analysis and data interpretation of clinically relevant molecular alterations in clinical practice. To evaluate the impact of preanalytical variables, we shipped *n* = 10 dedicated slides from an artificial FFPE reference standard that harbored the MET exon 14 skipping molecular alteration to *n* = 10 referral centers involved in molecular testing. Manual and automated nucleic acid extraction protocols were adopted in a comparable number of centers (6 vs. 4). Interestingly, molecular analysis was successfully carried out in all instances without any remarkable differences related to RNA extraction and purification approach. Regarding the molecular analysis of MET exon 14 skipping, 5 out of 10 (50.0%) centers equally adopted RT-PCR and NGS, respectively. Interestingly, no technically relevant issues were observed among the institutions in the detection of MET exon 14 skipping. This point underlines that both applied technical strategies were able to detect MET exon 14 skipping by trained personnel. The predictive role of MET exon 14 skipping for treatment with target drugs has been widely demonstrated. In fact, in prospective clinical trials, tepotinib, capmatinib, and savolitinib showed high activity in advanced stage lung cancer patients harboring the MET exon 14 skipping mutation, including treatment-naïve patients and those affected by pulmonary sarcomatoid carcinoma [13,20]. Interestingly, in the VISION study, MET exon 14 skipping was assessed either in tissue biopsy, plasma samples, or both. However, a major issue in the detection of MET exon 14 skipping mutations is related to the heterogeneous nature of the alterations that can lead to this genomic aberration. Thus, the different approaches that can be adopted in the detection of these molecular events is an open issue. Overall, several different genomic tests have been employed to detect MET exon 14 skipping molecular alterations, including single-gene tests [14,16,21,22,23,24]. RT-PCR is among the most sensitive techniques available for mRNA detection and quantitation [25]. Overall, RT-PCR is an efficient, reliable, and cost-effective technique for the detection of the MET exon 14 skipping molecular alteration because it can target the E13/E15 splice region [13,25,26]. However, a major limitation in the adoption of this approach is that a limited RNA quality can lead to false negative results [24]. Thus, RT-PCR may be used as a valid screening approach to test patients harboring the MET exon 14 skipping mutation prior to confirming molecular data with NGS [16]. This latter approach is considered the gold standard for MET exon 14 skipping detection. Considering amplicon-based approaches, the superiority of RNA-based versus DNA-based approaches has been demonstrated. This phenomenon is related to the possibility of allele dropout that may occur if there is a single-nucleotide variant, a short indel in the primer region, or primer binding sites are missed in the case of deletion of an entire genomic region [27]. For example, it has been shown that in seven DNA-based amplicon NGS assays, none of the adopted panels were able to identify more than 63% of known MET exon 14 skipping molecular alterations [27]. Overall, the Ion AmpliSeq Colon and Lung Cancer Research Panel v2, improved with three additional amplicons to cover exon 14 and its surrounding introns, enabled an increased detection rate of MET exon 14 skipping molecular alterations [27]. The hybrid capture approach should be preferred over amplicon-based approaches due to the possibility of avoiding allele dropout [14,23]. RNA-based sequencing approaches may be preferable to DNA-based approaches in MET exon 14 skipping detection. In fact, the RNA-based approach demonstrated its superiority in a study by Davies et al., in which RNA- and DNA-based NGS assays had detection rates of 4.3% and 1.3%, respectively [15]. This phenomenon is related to the possibility of detecting the direct result of alterations leading to MET exon 14 skipping, represented by the fusion of exons 13 and 15 [15]. In addition, this approach may detect non-canonical intronic mutations that can determine splicing [28]. However, the main limitation of this approach is associated with the low quality of RNA extracted from FFPE samples. Accordingly, the implementation of a fully automatized NGS-based system may solve this issue [29]. It was widely demonstrated that an NGS system can reduce the failure rate test in the analysis of “scant” diagnostic tissue specimens [30]. In particular, a fully automated NGS system represents a highly reliable approach able to reduce inadequate testing rate, inter-laboratory variability, and turnaround time (TAT) [31].

## 5. Conclusions

The use of artificial controls was a useful tool to evaluate inter- and intra-laboratory reproducibility, as was previously demonstrated in experiments carried out by the Molecular Cytopathology Meeting Group. In this study, the adoption of an engineered cell line harboring a known genomic alteration that could be distributed to different laboratories was fundamental in the validation of molecular platforms not only for DNA-based but also for RNA-based biomarkers [17,18,19]. Further investigation is warranted to assess the impact of analytical workflow on real-world clinical samples in order to evaluate the critical points found in the MET exon 14 skipping analysis for selecting NSCLC patients for targeted therapy.

## Figures and Tables

**Figure 1 diagnostics-13-00629-f001:**
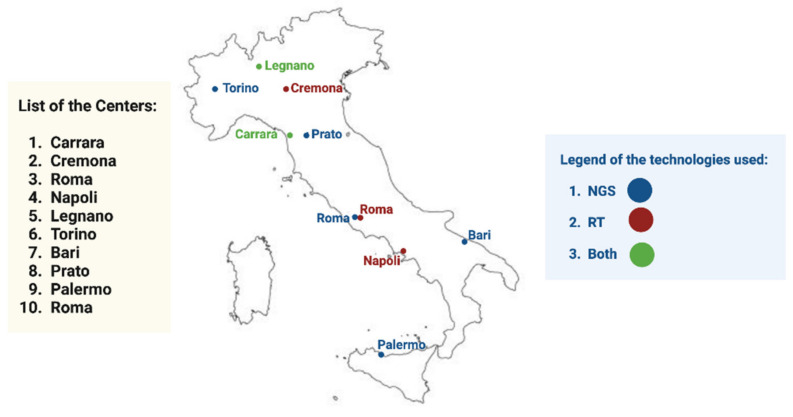
List of participating centers in this study. The techniques used in each center are color coded.

**Figure 2 diagnostics-13-00629-f002:**
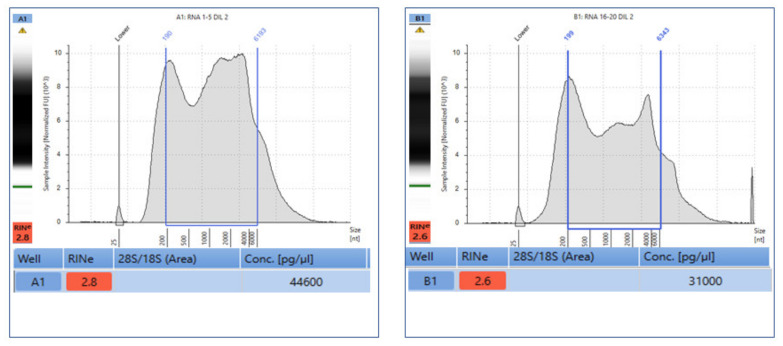
Schematic representation of RNA quantification and qualification from two representative slide sets of FFPE reference standard sample developed by Seracare Life Sciences on Tape Station 4200 (Agilent) platform.

**Figure 3 diagnostics-13-00629-f003:**
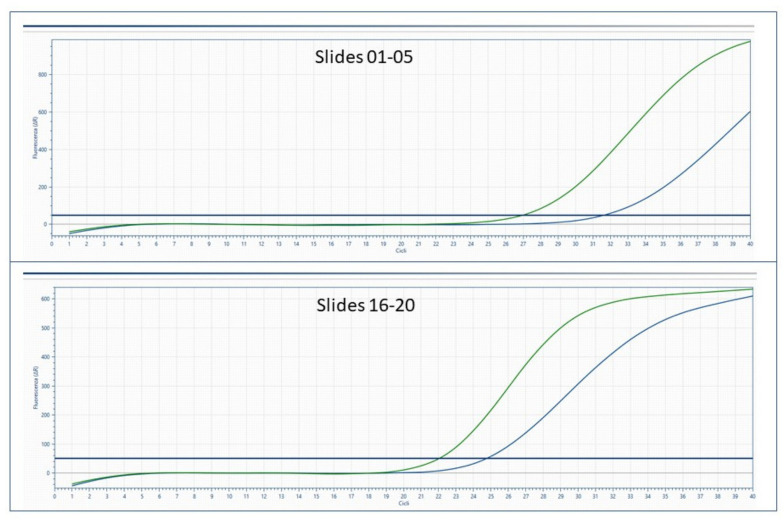
EasyPGX results for MET exon 14 skipping molecular alteration detected in processed slide sets.

**Table 1 diagnostics-13-00629-t001:** RNA-based analysis on reference standard samples.

*RT-PCR-Based Technologies*						
Center	Extraction Kit	RNA Amount (ng/µL)	Platform	Assay	Results	Cq Value
1	MagCore Total RNA FFPE One-step kit	NA *	EasyPGX	EasyPGX ready ALK/ROS1/RET/MET	METΔ exon 14	30.7
2	MagCore Total RNA FFPE One-step kit	10.0	EasyPGX	EasyPGX ready ALK/ROS1/RET/MET	METΔ exon 14	31.4
3	HGH Pure MIRNA isolation kit	19.5	EasyPGX	EasyPGX ready ALK/ROS1/RET/MET	METΔ exon 14	27.1
4	RNeasy FFPE Kit	40.7	EasyPGX	EasyPGX ready ALK/ROS1/RET/MET	METΔ exon 14	29.7
5	Maxwell RSC RNA FFPE Kit	8.7	EasyPGX	EasyPGX ready ALK/ROS1/RET/MET	METΔ exon 14	32.0
** *NGS-based technologies* **						
**Center**	**Extraction kit**	**RNA amount (ng/µL)**	**Platform**	**Assay**	**Results**	**Read count**
6	Maxwell RSC RNA FFPE Kit	4.3	iSeql100™	Myriapod^®^NGS Cancer Panel RNA	METΔ exon 14	160
7	Magmax FFPE DNA/RNA ultra kit	18.0	S5™	Oncomine Focus Assay	METΔ exon 14	NR **
8	Rneasy FFPE Kit	51.3	MiSeq	Myriapod^®^NGS Cancer Panel RNA	METΔ exon 14	176
9	Rneasy FFPE Kit	18.3	S5™	Oncomine Focus Assay	METΔ exon 14	7526
10	Rneasy FFPE Kit	49.0	Genexus	Oncomine Precision Assay	METΔ exon 14	2194

NA * (Not Available). NR ** (Not Reported).

## Data Availability

Not applicable.

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
