# Peer review of "An Italian Multicenter Perspective Harmonization Trial for the Assessment of MET Exon 14 Skipping Mutations in Standard Reference Samples"

_diagnostics, 2023, doi:10.3390/diagnostics13040629_

Round 1

Reviewer 1 Report

Concordance of MET exon-skipping detection among laboratories were elaborately conducted with convincing results and conclusions. However, to be published in the Journal of Diagnostics, I expect more work to be done and presented. For instance, how far the detection limit / sensitivity could reach using same cell line block with MET exon-skipping by different diluted cell line suspensions; what the performance of MET exon-skipping detection will be using biopsies with limited tumor content. A full story line from cell line to clinical samples should be constructed in order to be published in the Journal of Diagnostics

Author Response

Concordance of MET exon-skipping detection among laboratories were elaborately conducted with convincing results and conclusions. However, to be published in the Journal of Diagnostics, I expect more work to be done and presented. For instance, how far the detection limit / sensitivity could reach using same cell line block with MET exon-skipping by different diluted cell line suspensions; what the performance of MET exon-skipping detection will be using biopsies with limited tumor content. A full story line from cell line to clinical samples should be constructed in order to be published in the Journal of Diagnostics

We thank the reviewer for the suggestion. We have implemented the latest version of the manuscript with a paragraph in the results section and two additional figures able to elucidate the validation step of artificial sample purchased by Seracare Life Sciences.

Reviewer 2 Report

MET exon 14 skipping alterations occur in about 3 – 4% of advanced stage NSCLC patients and  is required for the molecular analysis for the clinical stratification of non-small cell lung cancer (NSCLC) patients.

Different technical approaches are available to detect MET exon 14 skipping in routine practice. The article compared the technical performance and reproducibility of MET exon 14 skipping testing strategies carried out in various centers were evaluated.

Each institution received a set of n=10 customized artificial formalin-fixed paraffin-embedded (FFPE) cell line  that harbored MET exon 14 skipping previously validated .

Each participating institution managed the reference slides according to internal routine workflow. MET exon 14 skipping was successfully detected by all participating institutions.

The article should be accepted as it nicely highlights that using artificial reference slides represent a valid tool and the importance of harmonization of  technical workflows for clinical testing that is s in routine practice. 

Minor comments; 

Remove lines 175-177

Analysis software version should be indicated for the NGS 

Author Response

The article should be accepted as it nicely highlights that using artificial reference slides represent a valid tool and the importance of harmonization of  technical workflows for clinical testing that is s in routine practice. 

Minor comments; 

Remove lines 175-177

Analysis software version should be indicated for the NGS

We thank the reviewer for the suggestion. We have modified the latest version of the manuscript accordingly.